CD4+CD25+ regulatory T cell therapy in neurological autoimmune diseases

Yuan Guobin 1
Liu Ying 2
Wang Hongquan 3
Yang Tingting 1
Liu Guangzhi guangzhi2002@hotmail.com 1
1 Department of Neurology, Beijing Anzhen Hospital, Capital Medical University , Beijing , China
2 Department of Neurology, Beijing Shunyi Hospital , Beijing , China
3 Department of Geriatrics, Aerospace Center Hospital, Peking University Aerospace School of Clinical Medicine , Beijing , China
Wu Hao
Electronic publication date: 2025 Jun 12
Publication date: 2025
Volume: 13
Electronic Location ID: e19450
Received 2024 Jun 24; Accepted 2025 Apr 21
Copyright: ©2025 Yuan et al.
Copyright year: 2025
Copyright holder: Yuan et al.
License: This is an open access article distributed under the terms of the Creative Commons Attribution License, which permits unrestricted use, distribution, reproduction and adaptation in any medium and for any purpose provided that it is properly attributed. For attribution, the original author(s), title, publication source (PeerJ) and either DOI or URL of the article must be cited.
License URL: https://creativecommons.org/licenses/by/4.0/

Keywords: Regulatory T cell, Autoimmune diseases, Nervous system, Neurological autoimmune diseases, Cell therapy

Funding: Beijing Natural Science Foundation Program Scientific Research Key Program of Beijing Municipal Commission of Education KZ201910025030 This work was supported by the Beijing Natural Science Foundation Program and Scientific Research Key Program of Beijing Municipal Commission of Education (No. KZ201910025030). The funders had no role in study design, data collection and analysis, decision to publish, or preparation of the manuscript.

==============================
CD4+CD25+ regulatory T cells (Tregs) play a critical role in maintaining immune tolerance. They are essential for the initiation and progression of autoimmune diseases affecting the nervous system. Recently, the correlation between Tregs and neurological autoimmune diseases, as well as their therapeutic potential, has become a central focus of research. Currently, various methods for in vivo or in vitro generation and expansion of CD4+CD25+ Tregs are under investigation; however, their application in cellular therapy is anticipated to face additional challenges. This article primarily delves into the development and function of CD4+CD25+ Tregs, the role of Tregs in neurological autoimmune disease pathology, basic methods for enhancing therapies, and recent advancements and challenges in cellular therapy for neurological autoimmune diseases.

Introduction

Neurological autoimmune diseases (NADs) are a heterogeneous group of conditions affecting the central or peripheral nervous system and are characterized by an inappropriate immune response that mistakenly targets the nervous system (Iranzo, 2020; Sweeney, 2018). NAD mainly affects the central nervous system (CNS), including autoimmune encephalitis, multiple sclerosis (MS), and neuromyelitis optica spectrum disorders (NMOSD), and the peripheral nervous system (PNS), including Guillain-Barré syndrome (GBS), neuromuscular junctions such as myasthenia gravis (MG), and muscles such as polymyositis. Treatment of NAD is composed mostly of broad-acting immunomodulators (i.e., intravenous immunoglobulin, interferon-β (IFN-β), anti-CD20 monoclonal antibodies, etc.) and immunosuppressants (i.e., corticosteroids, tacrolimus, and azathioprine) to restore normal immune function. Because most of NADs are not curative, lifelong administration are required and may cause various side effects, including alopecia, bone marrow suppression, and neuropsychiatric manifestations (Biolato et al., 2021; Briani & Visentin, 2022; Goeb et al., 2006; Minami et al., 2011).

Regulatory T cells (Tregs) play a crucial role in the maintenance of peripheral immune tolerance and homeostasis, prevention of autoimmunity, and control of chronic inflammatory diseases (Pisetsky, 2023). This T-lymphocyte subpopulation develops in the thymus and peripheral tissues of the immune system (Savage, Klawon & Miller, 2020). In addition to CD4+CD25+ Tregs, these types of immunosuppressive T cells include T helper type 3 (Th3) cells (characterized by CD4+CD25−phenotype), T-regulatory 1 (Tr1) cells (characterized by CD4+CD25low phenotype), CD8+ Tregs, natural killer T cells (NKTs), regulatory γδ T cells and other various subtypes (Bozward et al., 2021; Catalán et al., 2021; Ferreira et al., 2019; Mishra et al., 2021; Peters, Kabelitz & Wesch, 2018). Although different subtypes have unique features, Tregs is indispensable for inducing immune tolerance via two main mechanisms: bystander suppression and infection tolerance (Ferreira et al., 2019; Pascual et al., 2014).

To date, numerous studies have been conducted on CD4+CD25+ Tregs, which are characterized by low proliferative capacity and high expression of an epigenetically stabilized transcription factor, forkhead box protein 3 (Foxp3) (Sakaguchi et al., 2008). Foxp3 is a key transcription factor critical for maintaining the immunosuppressive function of Tregs (Golzari-Sorkheh & Zúñiga-Pflücker , 2023). Compared to other Treg subsets, CD4+CD25+ Tregs exhibit higher Foxp3 expression levels, thereby demonstrating enhanced therapeutic efficacy in autoimmune diseases. This type of CD4+CD25+Foxp3+ Tregs has been extensively studied (Amini et al., 2022; Jeyamogan et al., 2023; Wu et al., 2019; Zhao, Liao & Kang, 2017). Tregs mediate their tolerogenic effects through multiple modes of action, including production of inhibitory cytokines, cytokine starvation of effector T cells (Teffs) (e.g., interleukin [IL]-2), Teff suppression by metabolic disruption, and modulation of antigen-presenting cells (APCs) maturation or function (Kanamori et al., 2016; Shevach, 2018). Together, these activities result in broad control of various immune cell subsets, leading to the suppression of cell activation/expansion and effector functions. Moreover, these cells can facilitate tissue repair to complement their suppressive effects (Delacher et al., 2021; Moon et al., 2023). In recent years, there has been an effort to use CD4+CD25+ Tregs as a new therapeutic approach to treat autoimmune and other immunological diseases (Amini et al., 2022; Bluestone et al., 2023; Grover, Goel & Greene, 2021; Zhao, Liao & Kang, 2017).

Currently, cultivation of CD4+CD25+ Tregs with robust proliferation and sustained functionality is a major concern. Polyclonal and antigen-specific Tregs are generally produced either in vivo or ex vivo induction (Eggenhuizen, Ng & Ooi, 2020; Masteller, Tang & Bluestone, 2006; Selck & Dominguez-Villar, 2021). The methodology for generating polyclonal Tregs has reached a relatively advanced stage of development. However, the potential treatment efficacy of polyclonal Tregs infusion is related to bystander immunosuppression and may compromise normal defense functions, with increased risks of general immunosuppression and associated adverse reactions (Balcerek et al., 2021; Pilat & Sprent, 2020). Mounting evidence from animal studies proves higher efficacy of antigen-specific Tregs in regulating pathological immune responses in a disease-specific fashion, probably because the infused Tregs migrate into tissues expressing the cognate antigen, contributing to the stronger localized regulation of inflammation (LaMothe et al., 2018; Sun et al., 2018). Consequently, antigen-specific Tregs infusion is becoming a promising strategy for prevention or management of chronic inflammation that underlies various autoimmune disorders (Sakaguchi, Kawakami & Mikami, 2023; Selck & Dominguez-Villar, 2021).

In this review, we summarize the development and function of Tregs, the role of Tregs in NAD pathology, basic methods of Treg therapy, and recent advancements and challenges in cellular therapy for neurological autoimmune diseases. This review aims to elucidate the current state of Treg therapy for neuroscientists or bioengineers, to highlight additional challenges, and to offer innovative perspectives for neurologists in the treatment of neurological autoimmune diseases. It posits that, in the future, more refined Treg therapy techniques could be applied in this field, thereby benefiting patients.

Survey Methodology

We selected 194 articles through an exhaustive search of the PubMed online database using these subject terms including Treg (or CD4+CD25+ Treg) and/or neurological autoimmune diseases (MS, MG, GBS and NMOSD) and/or therapy, simultaneously excluding publications of inferior quality or redundant content. Furthermore, we conducted a careful synthesis and analysis of all sourced literature, ensuring that this review maintains a high level of comprehensiveness, accuracy and timeliness.

Development, Migration and Residency of CD4+CD25+ Tregs

Development of CD4+CD25+Tregs

A population of thymus-derived suppressor T lymphocytes were discovered more than 50 years ago (Gershon & Kondo, 1971; Nishizuka & Sakakura, 1969). Subsequently, an increasing number of studies have been conducted on this cell population (Chen et al., 1994; Sakaguchi et al., 1985). Sakaguchi et al. (1995) first identified the specific surface expression of CD25 (IL-2 receptor [IL-2R] alpha-chain) molecules on CD4+ T cells in the peripheral blood of mice. Upon elimination of CD4+CD25+ cells, the general immunosuppression is relieved, thereby accelerating immune responses to non-self-antigens, and causing autoimmune responses to certain self-antigens. Decades of research have extensively explored the phenotypes and regulation of Tregs (Fig. 1).

Figure 1 Timeline diagram depicting essential discoveries in the research field of CD4+CD25+ Treg.

Based on their developmental origin, Tregs can be further classified into thymus-derived Tregs (tTregs) and peripherally derived Tregs (pTregs) (Martin-Moreno, Tripathi & Chandraker, 2018; Robertson et al., 2019). Others divide Tregs into natural Tregs (nTregs) and induced Tregs (iTregs) based on the characteristic of natural occurrence (Edinger, 2009; Schmitt & Williams, 2013). However, others argue that there are three subgroups of Tregs: nTregs, pTregs, and iTregs (Göschl, Scheinecker & Bonelli, 2019; Zhang, Guo & Jia, 2021). nTregs and tTregs are fundamentally the same CD4+ T cells subpopulations, undergoing natural development and maturation within the thymus. Mature CD4+CD25+ Tregs can also be induced from naïve T cells by TGF-β and IL-2 stimulation; the resulting cells are termed iTregs or pTregs when generated in vitro or in vivo. Unlike nTregs, iTregs have been demonstrated to be unstable, and approaches to generate stable iTregs have been developed for clinical utility (Fig. 2) (Kanamori et al., 2016; Pacholczyk, Kraj & Ignatowicz, 2002).

Figure 2 The process of development and differentiation of CD4+CD25+ Tregs.

nTregs originate from the thymus. Their development and maturation depend on the synergistic stimulation of TCR and CD28. Foxp3 is induced to be highly expressed during the thymic development of nTregs. iTregs are mainly divided into Foxp3+ and Foxp3− Tregs. The former are generated from naïve T cells under the stimulation of cytokines. When induced in vitro, they are referred to as iTregs, whereas when generated in vivo (outside the thymus, such as in the gut or peripheral blood), they are known as pTregs. The latter are primarily categorized as Th3 cells and Tr1 cells.IL-2, in conjunction with TGF-β, can induce naïve CD4+CD25− T cells to transform into CD4+CD25+ T cells and express Foxp3. However, when naïve CD4+ T cells receive signals from IL-6, the functionality of Foxp3 is suppressed, inducing differentiation towards Th17 cells. Tregs, regulatory T cells; nTregs, natural Tregs; iTregs, induced Tregs; pTregs, peripherally induced Tregs; TCR, T cell receptor; Foxp3, forkhead box protein 3; Th, helper T cell; Tr, T-regulatory; IL, interleukin; TGF-β, transforming growth factor-β (created with BioRender.com).

Differentiation and functional expression of Tregs are regulated by a multitude of cytokines. T helper type 17 (Th17) cells originate from the same subset of CD4+ T cells. Initially, precursor Th (Th0) cells differentiate into intermediate cells. The subsequent differentiation trajectory primarily depends on the type of cytokines present in the microenvironment of the organism. Th17 cells and Tregs play opposing roles. The former hinders immune tolerance, whereas the latter induces it. A dynamic imbalance between them is pivotal for the onset of autoimmune disorders (Astier & Kofler, 2023; Lee, 2018; Yasuda, Takeuchi & Hirota, 2019).

IL-6 plays a significant role in modulating the equilibrium between Th17 cells and Tregs. It collaboratively induces the differentiation of naïve CD4+ T cells into Th17 cells with TGF-β, while suppressing the differentiation of TGF-β-induced Tregs (Kimura & Kishimoto, 2010).

Foxp3, presently the most discerning molecular marker of Tregs, participates in the differentiation and functional regulation of Tregs. Foxp3 is expressed in the thymic development of nTregs, or highly expressed in iTregs induced by TGF-β and retinoic acid (Ilnicka et al., 2019; Schmidt et al., 2016; Sun, Yi & O’Connell, 2010). In the absence of cytokines, Foxp3 can suppress the function of the retinoic acid receptor-related orphan receptor (ROR) γt and promote Treg differentiation. However, when naïve T cells receive signals from cytokines such as IL-6, the function of Foxp3 is inhibited, inducing the differentiation of Th17 cells (Deng et al., 2019; Zeng et al., 2023; Ziegler & Buckner, 2009). Therefore, the expression and regulatory levels of Foxp3 and RORγt determine the differentiation direction of Th17 cells/Tregs, resulting in the development of autoimmune diseases (Lee, 2018; Zeng et al., 2023). Additionally, in orchestrating the differentiation and function of Tregs, Foxp3 is influenced by various protein complexes with which it interacts. Prior to initiating Foxp3 transcription, Tregs have already been identified with homologous antigens and received T cell receptor (TCR) signals, triggered by TCR-induced transcription factors (NFAT, AP-1, and NF-κB), inducing the interaction between Foxp3 and AML1/Runx1 and NFAT, thereby exerting regulatory effects (Grover, Goel & Greene, 2021; Long, Luo & Zhu, 2022; Ono, 2020).

Signal transducer and activator of transcription 5 (STAT5) is another pivotal factor involved in Treg differentiation and activation. Mounting evidence indicates that IL-2 binds to the IL-2R on the surface of CD4+CD25+ Tregs, thereby activating STAT5. This, in turn, leads to the binding of STAT5 to the Foxp3 promoter, ultimately facilitating Treg cell generation (Jones, Read & Oestreich, 2020; Li & Park, 2020; Mao et al., 2019; Paradowska-Gorycka et al., 2020). Consequently, IL-2 regulates the differentiation of CD4+CD25+Foxp3+ Tregs via the STAT5 pathway, concurrently suppressing Th1 and Th17 cells as well as their secretion of IFN-γ and IL-17. This may be crucial for the onset and progression of autoimmune diseases (Mao et al., 2019; Paradowska-Gorycka et al., 2020; Xie et al., 2023).

Thymic development of nTregs is contingent upon the synergistic stimulation of TCR and CD28. CD28 is indispensable for the peripheral homeostasis, expansion, and survival of nTregs (Kitazawa et al., 2009). The maturation of iTregs necessitates the production of IL-2 and TGF-β, rather than the cooperative stimulation with CD28 (Cassis, Aiello & Noris, 2005). Additionally, IL-2, in conjunction with TGF-β, has the capacity to jointly induce the conversion of initial CD4+CD25−T cells into CD4+CD25+ T cells, thereby expressing Foxp3 (Zheng et al., 2007). Figure 2 illustrates the specific Treg differentiation process.

Migration and residency of CD4+CD25+Tregs

Upon maturation in the thymus, CD4+CD25+ Tregs enter systemic circulation or lymphatic tissues, with a subset further migrating to various non-lymphoid tissues, where they form transiently resident tissue Tregs in small numbers (Burton et al., 2024). In contrast to lymphoid tissue Tregs, these non-lymphoid tissue Tregs typically exhibit common molecular phenotypes, including CD69, CD103, programmed cell death protein-1 (PD-1), and so on, which facilitate their short-term residence in tissues and enable them to perform specific function (Burton et al., 2024; Shou et al., 2021; Evrard et al., 2023; Pasciuto et al., 2020).

Unlike Tregs in other tissues, brain Tregs undergo a distinct migration and residence process. Under physiological conditions, only a small subset of Tregs can cross the blood-cerebrospinal fluid barrier (not the intact blood–brain barrier) to enter the brain parenchyma (Korin et al., 2017; Hrastelj et al., 2021). The low IL-2 microenvironment within the CNS further limits their long-term residence, typically resulting in their renewal after approximately three weeks (Burton et al., 2024). During inflammation, Tregs are activated in lymphoid organs and are directed by chemokine C-C motif ligands (CCL) such as CCL2 and CCL5. These cells upregulate chemokine C-C motif receptors (CCR) like CCR2 and CCR5, which allow them to recognize self-antigens within the blood–brain barrier (Liston et al., 2024; Ben-Yehuda et al., 2021). Consequently, Tregs migrate through capillary endothelial cells, the basal lamina, and other barriers, entering the perivascular space. Once within this space, Tregs express high levels of specific TCRs, which interact with major histocompatibility complex II (MHC II) molecules on CNS target cells (Liston et al., 2024; Liston, Dooley & Yshii, 2022). This interaction drives their passage through the astrocyte limitans, allowing them to enter the brain parenchyma and target the corresponding inflammatory regions within the CNS.

Moreover, compared to other non-brain tissue Tregs, brain Tregs exhibit distinct gene expression and protein secretion patterns, which are associated with their unique role in neural repair (as discussed in the next section).

Function of CD4+CD25+ Tregs

Treg therapy in NADs not only relies on its immunosuppressive function to eliminate inflammation but is also dependent on its neuroprotective and neural repair capacity, which is quite different from immunomodulation.

Tregs exert their immunosuppressive effects mainly through direct cellular interactions and the secretion of inhibitory cytokines. Additionally, they mediate immunosuppression through cell lysis, extracellular vesicles (EVs) release, and disruption of metabolic dysregulation (Figs. 3A–3E).

Figure 3 The mechanism of CD4+CD25+ Tregs exerting their immunosuppressive function.

(A) Tregs inhibit immune cell proliferation by directly binding to corresponding receptors (CD80/CD86 molecules, MHC-II molecules) on target cells through CTLA-4 and LAG3; (B) Tregs primarily achieve immunoregulation with a negative impact on DCs, Th1, Th17 cells, and others through the secretion of inhibitory cytokines, such as IL-10, IL-35, and TGF-β; (C) Tregs employ granzyme/perforin-mediated cytotoxic T cells for cellular lysis; (D) Tregs establish intercellular communication by releasing EVs, thereby modulating immune responses; (E) Tregs exhibit elevated affinity for IL-2 due to the expression of CD25 (IL-2 receptor), outcompeting other immune cells, ultimately leading to cellular apoptosis. Alternatively, Tregs, by expressing CD39 and CD73, facilitate the conversion of ATP to cAMP, subsequently transformed into adenosine. This adenosine binds to adenosine receptors on the surface of effector T cells, exerting immunosuppressive effects. Tregs, regulatory T cells; Teff, effector T cell; DCs, dendritic cells; MHC, major histocompatibility complex; CTLA-4, cytotoxic T lymphocyte-associated antigen-4; LAG3, lymphocyte-activation gene 3; Th, helper T cell; TGF-β, transforming growth factor-β; IL, interleukin; EVs, extracellular vesicles; ATP, adenosine triphosphate; cAMP, cyclic adenosine monophosphate (created with BioRender.com).

As critical T-cell costimulatory molecules constitutively expressed on the surface of Tregs, cytotoxic T lymphocyte-associated protein 4 (CTLA-4) competes with CD28, for CD80/CD86 on APCs and consequently leads to the suppression of conventional T cells (Tconv) cells (Lax et al., 2023; Qureshi et al., 2011). Furthermore, CTLA-4 upregulates indoleamine 2,3-dioxygenase (IDO), contributing to cell cycle arrest and elevated sensitivity to apoptosis in Teffs and dysregulated APCs (Cribbs et al., 2014; Massalska et al., 2022). Similar to CTLA-4, lymphocyte activation gene 3 (LAG-3) is also expressed on Tregs, and suppresses the function of dendritic cells (DCs) through binding to MHC-II molecules (Gagliani et al., 2013; Li et al., 2023). Certain chemotactic factors lead to the aggregation of Tregs around immune cells, exerting their effects through cell–cell contact (Müller et al., 2007; Xu et al., 2023).

Tregs primarily achieve immunomodulation by secreting inhibitory cytokines such as IL-10, IL-35, and TGF-β. TGF-β, beyond inducing Foxp3 expression and negatively regulating immune cells, also collaborates with IL-2 in the induction of CD4+ T cells transitioning into Tregs (Gu et al., 2022; Zheng et al., 2007). IL-35, which belongs to the IL-12 family of inhibitory cytokines, effectively suppresses T-cell proliferation and induces the generation of Tregs to dampen inflammatory responses (Collison et al., 2007; Ye et al., 2021). As a well-known immunosuppressive factor, the molecular mechanism underlying the immunomodulatory activity of IL-10 on APCs and T cells remains controversial (Bergmann et al., 2007; Moaaz et al., 2019).

Tregs can directly induce apoptosis of target cells via cell–cell contact, which is attributed to the release of cytotoxic factors (e.g., granzymes) (Cao et al., 2007; Gondek et al., 2005). Tregs exert inhibition on target cells through a mechanism involving granzyme-mediated cytolysis, utilizing granzyme/perforin-mediated cytotoxicity carried out by NKs and CD8+ cytotoxic T cells to eliminate viral infections, tumor cells, and foreign antigens, among others (Kim, 2010; Lieberman, 2003; Sanders et al., 2022). The EVs generated by Tregs represent a finely tuned intercellular exchange mechanism. Through the release of EVs, intercellular communication is facilitated, thereby orchestrating immune responses and establishing an environment conducive to immune tolerance (Lin, Guo & Jia, 2022; Rojas et al., 2020). Tregs barely secrete IL-2, but promote IL-2 starvation from their surroundings via their high-affinity IL-2 receptor (CD25), leading to cytokine deprivation-induced apoptosis of Teff cells (Barthlott et al., 2005; Pandiyan et al., 2007). Thus, Tregs can impede Teff activation via the IL-2/STAT5 pathway, inducing metabolic perturbations that ultimately lead to apoptosis (Jones, Read & Oestreich, 2020; Li & Park, 2020; Pandiyan et al., 2007). Treg cells express CD39 and CD73 and produce adenosine and cyclic adenosine monophosphate (cAMP); the former can upregulate the intracellular cAMP of Teff via adenosine receptor 2A, and thus disrupt their metabolism (Horenstein et al., 2013; Park et al., 2012; Xia et al., 2023).

Tregs within the CNS include brain-resident Tregs and brain Tregs recruited from peripheral circulation, accompanied by the occurrence of inflammation. Tregs have recently been proved to indirectly and directly regulate tissue repair, but their mechanisms remain unclear (Ali et al., 2017; Burzyn et al., 2013; Castiglioni et al., 2015). In many tissues, Treg are recruited to the injury site to prompt inflammation resolution and to control immunity after injury (Murphy et al., 2005). For example, Tregs can indirectly control regeneration by modulating neutrophils and helper T cells (Carbone et al., 2013; D’Alessio et al., 2009; Dombrowski et al., 2017; Weirather et al., 2014), inducing macrophage polarization (Aurora et al., 2014; Lavine et al., 2014). Moreover, Tregs have been found to directly facilitate regeneration by locally activating progenitor cells (Ali et al., 2017; Castiglioni et al., 2015).

The most prominent role of Tregs in the CNS is to induce myelin regeneration (Dela Vega Gallardo et al., 2019). Tregs act on oligodendrocyte progenitor cells (OPCs) by secreting cellular communication network-3 (CCN3) protein, accelerating their differentiation into oligodendrocytes, and promoting myelination (Dombrowski et al., 2017). In addition, brain Tregs express specific genes such as serotonin receptor 7 (Htr7), which encodes the serotonin receptor 5-HT7 and exerts a neuroprotective effect (Lenglet et al., 2002). Tregs inhibit the proliferation of neurotoxic astrocytes and mediate tissue repair by secreting amphiregulin (AREG), a ligand of the epidermal growth factor receptor (EGFR) (Ito et al., 2019). These Tregs alter the phenotype of microglia into neuroprotective and neurorepaired states by regulating brain-derived neurotrophic factor (BDNF) and osteopontin (OPN) (Liston, Dooley & Yshii, 2022; Shi et al., 2021). A recent study by Wang et al. (2023) revealed that Tregs regulated neuroinflammation and microglia pyroptosis, reducing demyelination, as well as promoting regeneration through TLR4/MyD88/NF-κB pathway. Further research is warranted to determine the potential functions of Tregs in CNS.

The role of CD4+CD25+Tregs in NAD pathology

Multiple sclerosis

MS is an inflammatory demyelinating disease occurring within the CNS. There are concerns that CD4+CD25+ Tregs may participate in the pathogenic process of MS through the induction or modulation of immune responses within the CNS (Dendrou, Fugger & Friese, 2015; Bhise & Dhib-Jalbut, 2016). Previous studies have revealed a significant decrease in the effector function of CD4+CD25+ Tregs in peripheral blood of MS patients compared to normal controls (Tarighi et al., 2023; Haas et al., 2005). Following immunotherapies such as IFN-β or anti-CD52 monoclonal antibodies, there is a marked increase in both the quantity and functionality of these cells (Chiarini et al., 2020; Kiapour et al., 2022), indicating an immunological deficiency of Tregs in MS. The dysfunction and reduced quantity of Tregs in MS attenuate the inhibition against other pro-inflammatory immune cells (such as CD4+ effector T cells, CD8+ T cells, Th1, Th17 cells, B cells, etc.), and the neural repair function of myelin slacken off, which further leads to the progression of neuroinflammation and demyelination. In addition, Tregs expressed certain chemokines or their receptors (such as CCL17 or C-X-C motif chemokine receptor 3 (CXCR3)) are recruited into neuroinflammatory sites of brain, inhibiting inflammatory response (Yang et al., 2015; Moreno Ayala et al., 2023). However, due to the migratory receptor dysregulation of Tregs in MS patients, migration to the brain is restricted and Tregs cannot serves various purposes in alleviating MS pathology (Danikowski, Jayaraman & Prabhakar, 2017).

Myasthenia gravis

MG is an organ-specific autoimmune disease, primarily elicited by highly specific autoantibodies directed against skeletal muscle acetylcholine receptors (AChR). Currently, the thymus is believed to play a pivotal role in its pathogenesis (Thapa & Farber, 2019). Within the thymus, a substantial population of activated autoreactive CD4+ T cells exists, particularly AChR-reactive CD4+ Tregs. However, abnormalities in the thymus of MG patients may lead to a deficiency or impairment in Treg function, subsequently resulting in an immunodynamic imbalance (Niu et al., 2020; Dong, 2021). A prior study indicated that the mRNA and protein expression of Foxp3 in peripheral blood CD4+CD25+CD127low Tregs of MG patients exhibit a decline compared with the healthy control group, resulting in impairment of their mediated inhibition on AChR-reactive T cells (Thiruppathi et al., 2012). This suggests that the functional deficiency of Tregs in MG patients may be correlated with a diminished expression of critical functional molecules such as Foxp3 (Kohler et al., 2017). Few studies have demonstrated that Tregs express dysfunctional migratory receptors in MG (Danikowski, Jayaraman & Prabhakar, 2017; Wei, Kryczek & Zou, 2006). Most research indicate that CD4+ follicular helper T cells expressing chemokine receptors CXCR5 involved in Tregs migration into germinal centers (Saito et al., 2005; Wen et al., 2016). However, the intensive mechanism of Tregs migration dysfunction is not yet fully clarified.

Guillain-Barré syndrome

GBS, characterized by demyelination of peripheral nerves and nerve roots along with vasculitic cellular infiltration, stands as an autoimmune peripheral neuropathy. Studies indicate a significant reduction in both the quantity and proportion of CD4+CD25+CD127− Tregs in the peripheral blood of GBS patients compared to the normal control group, suggesting immunological dysfunction (Wang et al., 2015; Súkeníková et al., 2024). Similarly, in comparison to the healthy control group, GBS patients (especially during the progressive or relapsing phases) exhibit a marked decrease in both the quantity and suppressive function of Tregs, along with reduced mRNA expression of Foxp3 in Tregs (Chi, Wang & Wang, 2008). In addition, GBS patients who receive intravenous immunoglobulin therapy experience a significant decrease in the frequency of Th1 and Th17 cells, along with an increase in the quantity of Tregs and enhanced suppression of effector T cell function (Maddur et al., 2014). This suggests the involvement of Th17/Treg cells in the pathogenesis of GBS. The functional deficiency of Tregs eventuate in an obvious abatement in the immunosuppressive function of effector T cells or B cells, which attack the self-myelin antigen of PNS, thereby promoting peripheral nerve inflammation. At present, there is no consensus on whether Tregs has the ability to promote peripheral nerve myelin regeneration, and more research is needed to explore.

Neuromyelitis optica spectrum disorders

NMOSD is an autoimmune, inflammatory, demyelinating disorder of the CNS. The pathogenic mechanisms of NMOSD are primarily influenced by B cell-mediated humoral immune regulation, with cellular immunity also playing a role (Murtagh et al., 2022). Only a few studies have shown that the percentage of CD4+CD25+Foxp3+ Tregs in peripheral blood T cells of acute phase of NMOSD patients is significantly lower than healthy controls. At the same time, animal experiments have found that the consumption of Tregs significantly enhances the loss of astrocytes and demyelination in the mice (Ma et al., 2021). The implications of these researches may herald an important role of Tregs abnormality in NMOSD pathogenesis. Paradoxically, Tregs primarily exert their immunomodulatory effects by influencing B cell humoral immunity, rather than directly participating in this process (Brill, Lavon & Vaknin-Dembinsky, 2019; Cao et al., 2023). Further researches will be needed on the specific role of Tregs in NMOSD.

CD4+CD25+Tregs enhancing therapies

Expanding of CD4+CD25+Tregsin vivo

Tregs can be selectively expanded in vivo using different methods, allowing for the polyclonal expansion of Tregs to mediate non-specific immune suppression. This in vivo expansion method is generally more straightforward than the adoptive Treg therapy. However, these methodologies often lack durability, specificity, and targeting precision, making it challenging to control the potential toxic side effects (Balcerek et al., 2021).

Several methodologies have been proposed for in vivo induction of Tregs (Fig. 4A). Initially, in the treatment of autoimmune diseases with anti-CD3 antibodies, CD4+ Treg expansion is induced, concomitant with the selective depletion of T-effector cells (Notley et al., 2010). Another approach to induce Tregs involves in vivo inhibition of mammalian target of rapamycin (mTOR) function. Administration of rapamycin, an mTOR antagonist, selectively augments the population of endogenous Tregs (Bagherpour et al., 2018; Zhao et al., 2022), concurrently demonstrating heightened immunosuppressive capabilities (Chen et al., 2010). Furthermore, a variety of studies have revealed that those expressing IL-2Rαβγ exhibit a higher affinity and sensitivity to IL-2 compared to conventional T cells expressing IL-2Rβγ (Jones, Read & Oestreich, 2020), thus low doses of IL-2 can stimulate the proliferation of Tregs (Dong et al., 2021; Graßhoff et al., 2021; Harris, Berdugo & Tree, 2023). Consequently, low-dose IL-2 therapy has been utilized as a new practical method to induce Treg expansion in vivo (Harris, Berdugo & Tree, 2023; Qiao et al., 2023; Zhou, 2022). However, implementing this approach is challenging. Given the small size of IL-2, it is quickly excreted in the urine, resulting in a relatively short half-life and diminished therapeutic sustainability. Moreover, as CD25 expression is not exclusive to Tregs and also at lower levels in activated T cells, low-dose IL-2 could stimulate the proliferation and activation of effector T cells (Harris, Berdugo & Tree, 2023; Qiao et al., 2023).

Figure 4 Different pathways of CD4+CD25+ Treg cell-based therapies.

(A) Administering anti-CD3 antibodies, rapamycin, and low-dose IL-2 can instigate in vivo expansion of CD4+CD25+ Tregs. (B) The GM-CSF-neuronal antigen T cell source vaccines along with biodegradable microparticle (MPs) can elicit the proliferation of antigen-specific Tregs in mice. (C) The human CD4+CD25+CD127low Tregs, isolated through flow cytometry, are induced and cultured ex vivo using CD3/CD28 beads and a low dose of IL-2 to yield functionally robust, polyclonal Tregs for adoptive transfer therapy. (D) Isolated human CD4+ T cells, when co-cultured with activated B cells and peptide segments, can be ex vivo induced to expend, yielding antigen-specific Tregs suitable for therapy; mice-derived antigen-specific Tregs engineered through TCR and CARs technologies can be manufactured for adoptive transfer therapy. Tregs, regulatory T cells; Ab, antibody; IL, interleukin; GM-CSF, granulocyte-macrophage colony stimulating factor; MPs, biodegradable microparticle; TCR, T cell receptor; CARs, chimeric antigen receptors (created with BioRender.com).

Current approaches for inducing antigen-specific Tregs in vivo rely primarily on the introduction of exogenous antigens into the body to stimulate and induce initial T cell responses. However, the safety of this method requires significant scrutiny. In recent years, microbial vaccines have garnered attention as granulocyte-macrophage colony-stimulating factor (GM-CSF)-neuronal antigen T-cell source vaccines, relying on the recognition of low-affinity T-cell antigen receptors, inducing the expansion and activation of CD4+CD25highFoxp3+ Tregs in myelin-specific TCR transgenic mice (Moorman et al., 2018). Furthermore, biomedical engineering for in vivo induction of Tregs has emerged as a prominent area of research. Rhodes et al. (2023) recently developed a novel method for inducing the expansion and activation of Tregs in vivo. They achieved this by designing new types of biodegradable bioengineered particle-biodegradable microparticles (MP) to induce immune tolerance. A schematic of this process is shown in Fig. 4B.

Adoptive transfer therapy of CD4+CD25+Tregs

To date, ongoing clinical/preclinical trials for Treg therapies have been conducted, with the majority of treatment protocols using intravenous infusion of ex vivo-expanded autologous Tregs.

Methods for the ex vivo expansion of polyclonal Tregs have now reached a relatively mature and stable stage. This protocol began with the isolation of CD4+CD25highCD127low Tregs from the peripheral blood, followed by stimulation with CD3/CD28 magnetic beads in the presence of IL-2, collectively activating and proliferating Tregs (Fig. 4C) (Balcerek et al., 2021). This approach has successfully yielded a sufficient number of well-functioning Tregs in graft-versus-host and autoimmune disease patients, with some trials showing signs of disease amelioration (Brunstein et al., 2011; Chwojnicki et al., 2021; Marek-Trzonkowska et al., 2016).

Although the adoptive transfer of polyclonal Tregs to induce immune tolerance has been proven safe and effective, the current research emphasis lies in the development of antigen-specific Tregs. Various studies suggest that, in comparison to polyclonal Tregs, antigen-specific Tregs hold distinct advantages over polyclonal Tregs in more efficiently suppressing effector T cells and promoting immune tolerance (Sagoo et al., 2011; Selck & Dominguez-Villar, 2021). In the context of cell therapy through adoptive transfer, antigen-specific Tregs require fewer cells to induce immune tolerance and simultaneously reduce the side effects associated with broad immunosuppression.

Several studies have demonstrated that co-culturing sorted Tregs with APCs or B cells in vitro can induce the expansion of antigen-specific CD4+CD25highCD127low Tregs (Putnam et al., 2013). Previous studies have shown that naïve CD4+ T cells can be induced and expanded from the peripheral blood of patients with MS to yield myelin basic protein 85–99 (MBP85−99) specific CD4+CD25+ Tregs (Fig. 4D) (Xiang et al., 2016). Another method for generating antigen-specific Tregs in vitro involves the use of synthetic receptors, including engineered TCRs and chimeric antigen receptors (CARs), to alter the specificity of polyclonal Tregs. A variety of studies have revealed that the adoptive transfer of Tregs engineered with MBP-specific TCRs can effectively alleviate symptoms (Fig. 4D) (Kim et al., 2018). However, the clinical translation of Tregs manufactured using TCR engineering is, to some extent, constrained by MHC limitations. The development of CARs enables Tregs to recognize their antigens directly in a non-MHC-restricted fashion (Arjomandnejad, Kopec & Keeler, 2022; Sadelain, Brentjens & Rivière, 2013). A recent study demonstrated that myelin oligodendrocyte glycoprotein (MOG)-specific CAR-Tregs can, by binding to Foxp3 in effector T cells, enable MOG-specific Tregs to breach the blood–brain barrier and exert immunosuppressive effects within the CNS, thereby ameliorating disease symptoms (Fig. 4D) (Fransson et al., 2012).

CD4+CD25+Tregs therapy in NADs

Multiple sclerosis

Based on the pathogenesis and therapeutic principles of Tregs in MS, various studies have favored Treg-enhancing therapies. Current therapeutic strategies for MS encompass indirect Treg expansion and functional augmentation. A phase II randomized controlled trial (RCT) revealed that low-dose IL-2 therapy selectively activates Tregs, mitigates symptoms of relapsing-remitting MS (RRMS), and markedly decreases radiologically detectable gadolinium-enhancing lesions (Louapre et al., 2023). Pharmacologic modulation of Treg activity via small-molecule inhibitors or biologic agents represents a promising avenue for MS therapeutics. Nevertheless, such interventions carry substantial adverse effects, rendering the adoptive transfer therapy a critical focus for future MS research.

As is well known, the experimental autoimmune encephalomyelitis (EAE) models produce various phenotypes of human MS, universally accepted as the quintessential animal model for MS (Procaccini et al., 2015). Currently, the literature on adoptive transfer of polyclonal Tregs in the EAE model of MS is extensive. Kohm et al. (2002) reported that mouse CD4+CD25+ polyclonal Tregs reduced inflammatory cell infiltration in the spinal cord of EAE mice and inhibited the progression of EAE. Subsequent studies have confirmed the efficacy of polyclonal Tregs in improving EAE symptoms (McGeachy, Stephens & Anderton, 2005; Zhang et al., 2004). However, only a few clinical trials have been conducted on polyclonal Tregs in patients with MS. Only one Phase I clinical trial indicated that intravenous or intrathecal injections of ex vivo-expanded polyclonal Tregs led to symptom improvement, lesion reduction, and no significant adverse reactions in relapsing-remitting patients with MS (Chwojnicki et al., 2021).

Given the limitations of polyclonal Tregs, the preparation of antigen-specific Tregs and autologous transplantation for MS treatment have become current research hotspots. Several recent studies have shown that antigen-specific Tregs induced by intraventricular inoculation with microbe-based vaccines or engineered carrier particles can significantly improve EAE symptoms in mice. Moreover, histopathological examination showed a marked reduction in inflammatory infiltration compared with that in the control group (Moorman et al., 2018; Rhodes et al., 2023). To date, limited data are available regarding the cultivation of antigen-specific Tregs through in vitro induction (co-culture with APCs and specific peptide segments, TCR, and CAR engineering) for the treatment of EAE. Previous studies have indicated that, compared to polyclonal Tregs, ex vivo-induced mouse antigen-specific Tregs by MBP demonstrated significant efficacy in preventing symptom recurrence of EAE (Stephens, Malpass & Anderton, 2009). Furthermore, various studies have shown that functional and stable MBP antigen-specific Tregs can be cultured ex vivo from the peripheral blood of mice or patients with MS using TCR and CAR-T engineering techniques. When intravenously administered to EAE mice, these cells suppress the autoimmune pathology of EAE and significantly improve neurological function scores of EAE mice (DePaula Pohl et al., 2020; Fransson et al., 2012; Kim et al., 2018). However, no clinical trials of antigen-specific Tregs have been reported for the treatment of MS. Moreover, these animal experiments lack effective humanized models and produce less compelling experimental results, thus making it challenging to conduct further clinical trials in humans. Animal and clinical trials on adoptive Treg transfer therapy for MS are presented in Table 1.

Table 1 Preclinical and clinical studies demonstrating increased efficacy of adoptive Tregs therapies in neurological autoimmune diseases.

Pre-clinial studies	
Disease	Model	Treg population	Evidence of functional manifestation	Ref.	
MS	C57BL/6 mice	CD4+CD25+CD62Lhigh T cells from peripheral LN of mice	Significant protection from the clinical development of MOG35−55induced EAE compared to non-Treg (CD25-)	Kohm et al. (2002)	
MS	SJL/J, C57BL/6J and IL-10-deficient mice on a C57BL/6J background	CD4+CD25+ T cells from spleen and LN of mice	CD4+CD25+ Tregs suppress the immune responses of pathogenic T cells of PLP peptide-immunized EAE mice through secreting IL-10	Zhang et al. (2004)	
MS	C57BL/6 (Ly5.2+ and Ly5.1+) mice	CD4+CD25+ T cells from the peripheral LN of mice by co-culturing with irradiated splenic APC and anti-CD3	CNS-derived CD4+CD25+ T cells suppress induction of EAE	McGeachy, Stephens & Anderton (2005)	
MS	HLA-DR15 transgenic mice	Engineered TCR MBP-specific Tregs in vitro from MS patients	Amelioration of EAE symtoms in MOG-immunized DR15 transgenic mice and suppression of autoimmune pathology in EAE	Kim et al. (2018)	
MS	C57BL/6 mice	GFP/CARαMOG-Foxp3-engineered CD4+ T cells in vitro from the mice	Prominent inhibiton capacity in vitro and myelin recovery in mice treated with engineered Tregs compared to controls	Fransson et al. (2012)	
MS	B10.PL, B10.PL×SJL, transgenic mice	CD4+CD25+ T cells in vitro culture with anti-CD3/CD28 beads from TCR-transgenic mice	Tregs prevent disease relapse when given after the onset of clinical EAE (no effect with polyclonal Tregs)	Stephens, Malpass & Anderton (2009)	
MS	HLA-DR15 transgenic mice on a C57Bl/6 background	Human Tregs expressing functional single-chain chimeric antigen receptors (scFv CAR), targeting either MBP or MOG	Potent suppression ability of autoimmune pathology in EAE compared to OB2F3-TCR Tregs	DePaula Pohl et al. (2020)	
MG	Lewis rats 6–7 weeks of age	Ex vivo generated CD4+ Tregs with anti-CD3/ CD28 beads and IL-2 from spleens of naive rats	Inhibition the progression of EAMG and down-regulation of humoral AChR-specific responses	Aricha et al. (2008)	
MG	Rats	CD4+CD25+ Tregs in vitro culture with anti-CD3/ CD28 beads, TGF-β and IL-2 from healthy rats	Significant inhibitory effect on EAMG	Souroujon et al. (2012)	
MG	Lewis rats aged 8–10 weeks	CD4+CD25+ Tregs sorted by co-culturing with DCs from spleens of rats	Significant suppressive effect of EAMG by autologous Tregs	Aricha et al. (2016)	
GBS	Lewis rats	CD4+CD25+ Tregs with anti-CD3/CD28 beads, IL-2, TGF-β and rapamycin	Reducction of inflammatory cells infiltration in the sciatic nerve, as well as myelin and axonal damage of EAN	Wang, Cui & Qian (2018)	
GBS	Lewis rats	Alloantigen specific CD4+CD25+ Tregs by ex vivo activation of PNM and rIL-2	Strong inhibition effect on EAN	Tran et al. (2020)	
NMOSD	C57BL/6 mice	CD4+CD25+ Tregs from spleens of mice	Suppression of inflammatory response and promotion of neuroprotection and regeneration	Ma et al. (2021)	
Clinical studies	
Disease	Methods	Phases	NCT number and status	Ref.	
MS	autologous polyclonal expanded Treg	Phase 1	EudraCT: 2014-004320-22; Recruiting	Chwojnicki et al. (2021)	
Notes.

Abbreviations Treg regulatory T cell

MS multiple sclerosis

MG myasthenia gravis

GBS Guillain-Barré syndrome

NMOSD neuromyelitis optica spectrum disorders

LN lymph nodes

MOG myelin oligodendrocyte glycoprotein

EAE experimental autoimmune encephalomyelitis

IL interleukin

PLP proteolipid protein

APC antigen-presenting cell

CNS central nervous system

HLA human leukoyte antigen

TCR T cell receptor

MBP myelin basic protein

GFP green fluorescent protein

CAR chimeric antigen receptor

Foxp3 forkhead box protein 3

EAMG experimental autoimmune myasthenia gravis

AChR acetylcholine receptor

TGF-β transforming growth factor-β

DCs dendritic cells

PNM peripheral nerve myelin

EAN experimental autoimmune neuritis

NCT national clinical trial

Myasthenia gravis

Presently, the treatment options for MG primarily include the use of immunomodulators to indirectly enhance the biological activity of Tregs or regulate the dynamic balance between Th17 cells and Tregs. Administering of GM-CSF can augment the inhibitory function of Tregs, elevate Foxp3 expression, thereby exerting a negative regulatory effect (Sheng et al., 2006; Thiruppathi et al., 2012). In a rat model of MG, experimental autoimmune myasthenia gravis (EAMG), crosstalk occurs between the JAK2/STAT3 and AKT/mTOR pathways. JAK2 inhibitors can regulate the Th17 cells/Treg balance by inhibiting both signaling pathways, consequently contributing to the amelioration of EAMG symptoms (Lu et al., 2023). Furthermore, IFN-γ possesses the capacity to promote an increased proportion of CD4+CD25+ Tregs, enhance Foxp3 expression (Huang, Wang & Chi, 2015). Additionally, MG-related animal experiments have indicated that currently used immunosuppressive drugs, such as fingolimod (Luo et al., 2020) and certain traditional Chinese medicines (i.e., astilbin and artesunate) (Meng et al., 2018; Meng et al., 2016), primarily regulate the balance between Th17 cells and Treg for therapeutic efficacy, providing new potential therapeutic targets for MG immunotherapy.

Regarding the adoptive transfer therapy of Tregs, numerous animal experiments have consistently confirmed their outstanding therapeutic potential (Aricha et al., 2008; Aricha et al., 2016; Souroujon et al., 2012). Intraperitoneal injection of ex vivo-induced and cultured CD4+CD25+ Tregs from healthy rats into the EAMG significantly ameliorated the clinical symptoms, reduced acetylcholine receptor (AChR) antibody titers, and suppressed the quantity and functionality of Th1 and Th2 cells. Moreover, an 8-week follow-up revealed a markedly higher survival rate than that of controls (Souroujon et al., 2012). Intravenous injection of ex vivo-induced and expanded autologous polyclonal CD4+CD25+Foxp3+ Tregs into EAMG rats showed inhibitory effects on Teff cells proliferation and improved myasthenic symptoms. In parallel with this finding, patients with MG showed decreased AChR-specific antibody levels in the peripheral blood (Aricha et al., 2016). Since numerical animal experiments were initially employed to investigate polyclonal Tregs, the next step should be taken to determine the efficacy and safety of antigen-specific Treg therapies for EAMG. This will shed light on the fundamental basis for conducting clinical trials related to Treg therapy for MG. Preclinical studies on MG are presented in Table 1.

Guillain-Barré syndrome

To date, the modulation of Treg bioactivity in patients with GBS is primarily achieved through immunomodulators or pharmaceutical agents. In vitro, through the activation of anti-CD3 and anti-CD28 antibodies, IFN-γ induced CD4+CD25− T cells from patients with GBS to generate CD4+CD25+ Tregs with heightened functional activity (Huang et al., 2009). Experimental autoimmune neuritis (EAN) is the most effective animal model of GBS. Some studies have indicated that atorvastatin, dexamethasone, and triptolide augment the quantity and function of CD4+CD25+Foxp3+ Tregs, thereby ameliorating EAN symptoms (Fagone et al., 2018; Shao, Fan & Tang, 2023; Xu et al., 2014). In several studies on adoptive transfer therapy using Tregs for animal experiments of GBS, ex vivo-induced and expanded rat polyclonal CD4+CD25+ Tregs were intravenously infused into the EAN model, leading to a significant reduction in sciatic nerve inflammatory cell infiltration and marked alleviation of myelin sheath and axonal damage (Wang, Cui & Qian, 2018). Building upon this research, Tran et al. (2020) reported that induced and cultured activated rat antigen-specific Tregs from the peripheral nerve myelin sheath and recombinant IL-2 were injected into the EAN, and demonstrated a notable reduction in clinical paralysis symptoms and degree of weight loss, along with accelerated disease recovery. Various animal experiments have provided a robust foundation for subsequent clinical trials (Table 1).

Neuromyelitis optica spectrum disorders

Currently, studies on Tregs therapy for NMOSD are scarce. In an animal experiment, adoptive transfer of Tregs attenuated brain damage in a mouse NMOSD model; reduced macrophage, neutrophil, and T-cell infiltration; and suppressed inflammatory responses by regulating the functional status of macrophages/microglia and reducing chemokines and proinflammatory factors (Table 1) (Ma et al., 2021). The clinical potential of Treg therapy for NMOSD is worth investigating.

Side Effects and Challenges

The therapeutic potential of CD4+CD25+ Tregs in NADs is substantial; however, several critical challenges must be addressed, including stability, safety and target specificity. Notably, while certain clinical trials have demonstrated the favorable safety profile of allogeneic Treg therapy, concerns regarding immune-mediated rejection and limited cell survival necessitate the prioritization of autologous Treg therapy in current clinical practice (Hennessy et al., 2023; Giganti et al., 2021). Furthermore, inducing Tregs proliferation in vivo and administering adoptive transfers of polyclonal Tregs may result in significant side effects. The antigenic heterogeneity of polyclonal Tregs can cause broad and indiscriminate immunosuppression, heightening the risk of infections and tumor progression (MacDonald, Piret & Levings, 2019). Additionally, during ex vivo expansion, polyclonal Tregs are susceptible to contamination by other activated T cells, which may provoke non-specific immune activation or inflammatory responses, ultimately compromising therapeutic stability and increasing the likelihood of severe adverse effects (Balcerek et al., 2021).

Compared to polyclonal Tregs, antigen-specific Treg adoptive transfer therapy selectively targets disease-associated antigens, facilitating precise migration to inflammatory sites, while mitigating systemic immunosuppressive adverse effects. However, this strategy remains associated with several potential challenges and risks. The primary concern is the functional instability of Tregs, as dysregulated epigenetic modifications—such as diminished Foxp3 expression—during ex vivo expansion or within the inflammatory microenvironment may result in the loss of immunosuppressive function or even their conversion into pro-inflammatory effector T cells, thereby aggravating tissue damage (Ou et al., 2021; Raffin, Vo & Bluestone, 2020; Bluestone et al., 2023). An additional challenge lies in Treg cell homing and persistence. Following adoptive transfer, Tregs demonstrate limited migration to inflammatory sites, potentially due to inadequate chemokine receptor expression. Moreover, their transient residency and survival in target tissues—possibly attributable to insufficient levels of survival-dependent cytokines such as IL-2 and IL-7—result in a rapid decline in Treg numbers and a concomitant loss of function (Ferreira et al., 2019; Baron & Turnquist, 2023). Furthermore, gene-editing strategies for Tregs carry inherent risks, as off-target effects associated with CRISPR/Cas9 or CAR-T technologies may induce genetic mutations or unintended tumorigenesis (Selck & Dominguez-Villar, 2021; Asmamaw Mengstie et al., 2024). Finally, the clinical translation of Treg therapy remains impeded by manufacturing complexities and scale-up challenges, including high production costs, suboptimal purity, low expansion efficiency, inconsistent quality control standards, and the necessity for contingency plans to address potential Treg dysregulation (Bluestone et al., 2023; Ferreira et al., 2019; Baron & Turnquist, 2023; Mamo et al., 2022). Given the unique characteristics of the CNS, the structural integrity of the blood–brain barrier—characterized by tight junctions, low chemokine expression, and restricted adhesion molecule availability—impedes the infiltration of Tregs into inflamed CNS regions. Moreover, the impaired migratory capacity of Tregs, coupled with insufficient support from the CNS microenvironment and limited targeting capabilities, further constrains their clinical application in NADs (Verreycken, Baeten & Broux, 2022; Olson, Mosley & Gendelman, 2023). In summary, CD4+CD25+ Treg-based therapies targeting NADs continue to confront substantial implementation challenges. Nevertheless, through advanced engineering strategies and precise modulation, the realization of highly efficient and safe Treg therapy holds promise for advancing personalized immunotherapy.

Conclusions

In conclusion, naive CD4+ cells originating from the thymus or periphery undergo a series of processes to differentiate into mature CD4+CD25+CD127lowFoxp3high Tregs, inducing immunological tolerance. They exert negative immunomodulatory effects through various pathways, including direct cell-to-cell contact and the secretion of inhibitory cytokines. Simultaneously, they play a distinctive neuroprotective and reparative role in CNS related diseases. Numerous studies have confirmed that the onset and progression of various NADs associated with a decreased number or functional deficiency of Tregs; thus, in vivo or ex vivo induced Tregs for adoptive transfer therapy in autoimmune neurological disorders pave the way for a promising treatment against these disorders. To date, most studies on Tregs therapy for NADs (i.e., MS, MG, and GBS) have been conducted in preclinical animal trials. The challenge of obtaining a sufficient number of antigen-specific Tregs with stable functionality and prolonged persistence remains a significant obstacle to curing patients with NADs; therefore, issues regarding the safety of Tregs therapy warrant further in-depth investigation.

Additional Information and Declarations

Competing Interests

Author Contributions

Data Availability

The authors declare there are no competing interests.

Guobin Yuan conceived and designed the experiments, performed the experiments, analyzed the data, prepared figures and/or tables, authored or reviewed drafts of the article, and approved the final draft.

Ying Liu conceived and designed the experiments, performed the experiments, analyzed the data, prepared figures and/or tables, authored or reviewed drafts of the article, and approved the final draft.

Hongquan Wang conceived and designed the experiments, analyzed the data, prepared figures and/or tables, authored or reviewed drafts of the article, and approved the final draft.

Tingting Yang conceived and designed the experiments, authored or reviewed drafts of the article, and approved the final draft.

Guangzhi Liu conceived and designed the experiments, authored or reviewed drafts of the article, and approved the final draft.

The following information was supplied regarding data availability:

This article is a literature review.

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
