# Peer review of "CD4+CD25+ regulatory T cell therapy in neurological autoimmune diseases"

_PeerJ, doi:10.7717/peerj.19450_

## Round 0.1 · original submission · Minor Revisions

In this manuscript, the authors reviewed CD4+CD25+ regulatory T cell therapy in treating neurological autoimmune diseases, thus potentially benefiting future studies. Yet, some additional clarification/discussion would be really appreciated. Kindly see the details below and comments from our reviewers.

If possible, it is suggested to elaborate on “pathogenesis … principles of Tregs in MS” and other neurological autoimmune diseases, in order to support the conclusion stating that the “onset and progression of various NADs associated with a decreased number or functional deficiency of Tregs.”

It is further suggested to compare/summarize the differences in Treg development in the neurological system compared to others, especially considering the existence of the blood-brain barrier and Tregs recruited from peripheral circulation to the brain.

Lines 424-425 state that “obtaining a sufficient number of antigen-specific Tregs with stable functionality and an extended half-life remains a significant obstacle to curing patients with NAD.” Yet, I am not sure this point is fully discussed.

Some very minor formality/grammatical issues were noticed while reviewing:
(1) It is suggested to start a new paragraph in line 333.
(2) It is suggested to delete the word “However” in line 79 or change it to something similar in view of the sentence immediately before it.

Reviewer 1 ·

Basic reporting

no comment

Experimental design

no comment

Validity of the findings

no comment

Additional comments

1. The author mentioned in the abstract that the application of CD4+CD25+ Tregs cellular therapy is anticipated to face additional challenges. However, these challenges and possible solutions are not clearly addressed in the later content.

2. Although the abbreviations of diseases were defined in previous paragraphs, it would be better if the author could use their full names in the subtitles.

3. The author indicated that all Tregs can induce immune tolerance. However, numerous studies have been conducted on CD4+CD25+ Tregs (not other subtypes) (lines 46-60). Please discuss the advantages of CD4+CD25+ Tregs over other subtypes, especially from the aspect of disease treatment.

4. To make it clearer for the reader, when the authors discuss the “Development and function of CD4+CD25+ Tregs,” please clarify if “thymus-derived Tregs (tTregs)” are the same as “nTregs,” explain their relationship, and update Figure 2 accordingly.

5. In the section “Development and function of CD4+CD25+ Tregs,” it is recommended to rearrange some of the writing to make it easier to understand. Line 119 mentions Foxp3+ when talking about the classification of Tregs; however, Foxp3 is introduced in a later paragraph (line 134).

6. When the author discusses the application of CD4+CD25+ Tregs on different types of NADs, it would be a more complete discussion if the side effects were also addressed.

Reviewer 2 ·

Basic reporting

This review paper discuss a current topic that affects many patients and would be a recent addition to the literature as a comprehensive review. The language is easy to follow and professional. Literature search was done broadly. Figures and tables are simple and summarizing the basic concepts of the topic.

- I would suggest some references could be added regarding indirect therapeutic regulation of Tregs such as the following: Louapre C, Rosenzwajg M, Golse M, Roux A, Pitoiset F, Adda L, Tchitchek N, Papeix C, Maillart E, Ungureanu A, Charbonnier-Beaupel F, Galanaud D, Corvol JC, Vicaut E, Lubetzki C, Klatzmann D. A randomized double-blind placebo-controlled trial of low-dose interleukin-2 in relapsing-remitting multiple sclerosis. J Neurol. 2023 Sep;270(9):4403-4414. doi: 10.1007/s00415-023-11690-6. Epub 2023 May 28. PMID: 37245191.

Experimental design

Literature search was done broadly and properly.

Validity of the findings

The insights are important regarding possible treatment options for neurological autoimmune diseases. I have few comments.

- Could do the authors add observed/possible side effects of the investigated treatments on human subjects? And maybe give a comparison with the current treatment alternatives to make the readers understand the advantages/disadvantages of pursued innovative treatments.

- Could authors give insights regarding why there hasn't been many clinical trials initiated yet? Are there other issues beyond technical difficulties? It was mentioned briefly in the conclusion part but I'd suggest to expand this discussion within the manuscript.

---

## Round 0.2 · accepted · Accept

Many thanks for the authors’ amendments. Really appreciated. I am now fully confident that publication of this manuscript aligns with our journal’s goal and will benefit our readers.

Yet, while reading the manuscript, I noticed some very minor grammatical/clarity matters (detailed below). Some quick grammatical/formality touch-ups may be helpful.

Is “including” missing from line 38 and right before “Guillain-Barré syndrome (GBS)”?

By “intravenous” in line 40, does it mean “intravenous injection of”? Is “intravenous immunoglobulin” really necessary in view of anti-CD20 antibody mentioned in the same sentence?

The sentence starting in line 49 is a little confusing, saying “T cells include …. regulatory B cells”?

Kindly change “cell” in line 69 to “cells” if accurate.

In line 85, kindly change “migrates” to “migrate” if accurate.

In line 124, kindly correct the spelling of “underging.”

By the sentence in line 131, would we like to say “T helper type 17 (Th17) cells originate from the same subset of CD4+ T cells” for a better flow?

In line 146, is it accurate to say “the differentiation into Th17 cells”?

In line 152, is it accurate to change “identified” to “been identified with”?
Is it accurate to change “facilitates” in line 178 and “enables” in line 179 to their plural version?

The term “else tissue” in line 196 is a little confusing.

Kindly confirm that the term “AChR-reactive CD4+ Treg” spanning lines 297 and 298 is accurate.

·

Basic reporting

The manuscript "CD4+CD25+ regulatory T cell therapy in neurological autoimmune diseases" demonstrates clear, unambiguous, and professional English throughout. The authors have effectively improved the language quality in response to previous comments, including appropriate terminology refinements such as replacing "extended half-life" with "prolonged persistence" (line 587).
Literature references are comprehensive and up-to-date, providing sufficient field background and context. The authors have successfully incorporated additional relevant studies as suggested, notably those discussing indirect therapeutic regulation of Tregs (lines 415-423). With 194 articles selected through exhaustive PubMed database searching (as stated in the Survey methodology section), the reference list constitutes a thorough foundation for this review.
The article structure is well-organized. The figures effectively illustrate key concepts, particularly the mechanisms of Treg function (Figure 3) and differentiation pathways (Figure 2), which aid considerably in understanding complex immunological processes. Tables summarize preclinical and clinical studies appropriately.
This review demonstrates broad and cross-disciplinary interest, bridging immunology and neurology, and clearly falls within the scope of PeerJ. The timely nature of this review is justified by recent advances in Treg therapy for autoimmune diseases and the lack of a comprehensive review specifically focusing on neurological applications.
The Introduction adequately introduces the subject, clearly defining neurological autoimmune diseases and establishing the relevance of CD4+CD25+ Tregs in this context. It effectively communicates that the intended audience includes neuroscientists, bioengineers, and neurologists seeking to understand current and potential Treg therapies (lines 96-100). The motivation—to elucidate therapeutic options for patients with currently incurable conditions—is clearly articulated.

Experimental design

The article content aligns well with PeerJ's Aims and Scope, focusing on the intersection of immunology and neurology. The investigation is rigorous, as evidenced by the systematic approach to the literature review described in the Survey methodology section.
The authors have transparently described their methodology and approach to literature selection, including database search and quality assessment criteria (lines 103-107). This methodology allows for replication of the review process.
The Survey Methodology is consistent with comprehensive, unbiased coverage of the subject. The authors' selection of 194 articles through exhaustive PubMed database searching with specific subject terms provides appropriate breadth, while their exclusion of "publications of inferior quality or redundant content" indicates attention to quality control.
Sources are adequately cited throughout, with appropriate quotations or paraphrasing. The authors have been diligent in attributing ideas and findings to their original sources.
The review is organized logically into coherent sections and subsections. The progression from basic Treg biology (development, migration, function) to pathological roles in specific diseases to therapeutic approaches (in vivo expansion, adoptive transfer, disease-specific applications) to challenges creates a natural flow that enhances comprehension of this complex topic.

Validity of the findings

This manuscript will contribute to the literature by synthesizing current knowledge on Treg therapy, specifically for neurological autoimmune conditions. The conclusions are well-stated and appropriately limited to supporting results from the literature reviewed. The authors have maintained scientific integrity by acknowledging both the therapeutic potential and the significant challenges of Treg-based approaches.
Overall, this manuscript presents a well-developed and supported argument that meets the goals established in the Introduction. The authors have successfully addressed how Tregs develop, function, and can be therapeutically manipulated in neurological autoimmune diseases, fulfilling their stated aim to "elucidate the current state of Treg therapy" (line 97).
The Conclusion effectively identifies unresolved questions and future directions. The authors specifically note that "the challenge of obtaining a sufficient number of antigen-specific Tregs with stable functionality and prolonged persistence remains a significant obstacle" (lines 586-588) and that "issues regarding the safety of Tregs therapy warrant further in-depth investigation" (lines 588-589). These statements appropriately guide future research efforts.

Additional comments

This revised manuscript represents a scientifically robust review of CD4+CD25+ regulatory T cell therapy in neurological autoimmune diseases. The authors have demonstrated good responsiveness to previous reviews, substantially enhancing the manuscript's scientific depth and clinical relevance through targeted additions and revisions.
Particularly commendable are:
The newly added "Migration and residency of CD4+CD25+ Tregs" section, which elucidates the unique characteristics of brain-resident Tregs
The expanded discussion of Tregs' role in specific neurological autoimmune conditions
The comprehensive "Side effects and challenges" section balances therapeutic optimism with a realistic assessment of current limitations.

Reviewer 4 ·

Basic reporting

good

Experimental design

good

Validity of the findings

good

Additional comments

good